# Mitochondrial Sirtuins in Chronic Degenerative Diseases: New Metabolic Targets in Colorectal Cancer

**DOI:** 10.3390/ijms23063212

**Published:** 2022-03-16

**Authors:** Antonino Colloca, Anna Balestrieri, Camilla Anastasio, Maria Luisa Balestrieri, Nunzia D’Onofrio

**Affiliations:** 1Department of Precision Medicine, University of Campania Luigi Vanvitelli, Via L. de Crecchio 7, 80138 Naples, Italy; antonino.colloca@studenti.unicampania.it (A.C.); camilla.anastasio@unicampania.it (C.A.); nunzia.donofrio@unicampania.it (N.D.); 2Istituto Zooprofilattico Sperimentale del Mezzogiorno, U.O.C. Food Control and Food Safety, 80055 Portici, Italy; anna.balestrieri@izsmportici.it

**Keywords:** chronic degenerative diseases, colorectal cancer, SIRT2, SIRT3, SIRT4, SIRT5, mitochondria, metabolism

## Abstract

Sirtuins (SIRTs) are a family of class III histone deacetylases (HDACs) consisting of seven members, widely expressed in mammals. SIRTs mainly participate in metabolic homeostasis, DNA damage repair, cell survival, and differentiation, as well as other cancer-related biological processes. Growing evidence shows that SIRTs have pivotal roles in chronic degenerative diseases, including colorectal cancer (CRC), the third most frequent malignant disease worldwide. Metabolic alterations are gaining attention in the context of CRC development and progression, with mitochondrion representing a crucial point of complex and intricate molecular mechanisms. Mitochondrial SIRTs, SIRT2, SIRT3, SIRT4 and SIRT5, control mitochondrial homeostasis and dynamics. Here, we provide a comprehensive review on the latest advances on the role of mitochondrial SIRTs in the initiation, promotion and progression of CRC. A deeper understanding of the pathways by which mitochondrial SIRTs control CRC metabolism may provide new molecular targets for future innovative strategies for CRC prevention and therapy.

## 1. Introduction

The current era of cancer research is increasingly focused on the possibility to intervene at epigenetic level, given the unparalleled possible implication for clinical translation. Despite the advances in therapeutic technology, cancer remains the second most common cause of death worldwide. In the advanced stages of cancer, metastasis and drug resistance are the key issues that result in patient death. In this scenario, there is an urgent need to identify novel sensitive markers for cancer prevention and early detection, and even more for the prediction of chemotherapy or targeted therapy effectiveness.

Sirtuins (SIRTs), class III histone deacetylases able to catalyze a unique β-nicotinamide adenine dinucleotide (β-NAD+)-dependent Nε-acyl-lysine deacylation reaction on histone and non-histone protein substrates [1], represent an attractive tool for epigenetic intervention. These proteins are highly conserved and the seven isoforms in humans, from SIRT1 to SIRT7, are distinguishable based on their function and subcellular localization. SIRT1, SIRT6 and SIRT7 are predominantly localized in the nucleus, SIRT3, SIRT4 and SIRT5 reside within the mitochondria, while SIRT2, predominantly found in the cytoplasm, has been found also in the nucleus as well as in the mitochondria [1,2]. Sirtuins can be divided into four classes, I to IV: SIRT1, SIRT2 and SIRT3 belong to class I; SIRT4 to class II; SIRT5 to class III; and SIRT6 and SIRT7 to class IV [1]. By virtue of their multifaceted activity in many biological processes such as cell survival, senescence, proliferation, apoptosis, DNA repair, cell metabolism and calorie restriction, an explosion of research on this family of proteins is ongoing [3,4,5]. Sirtuins affect cellular homeostasis acting at an epigenetic level, modulating gene expression without altering DNA sequence, and regulating enzymatic activity, mainly acting as deacetylases. They also catalyze other reactions such as desuccinylation, deglutarylation and demalonylation [1]. These modifications, strictly influenced by the environment, are reversible and can even be transmitted to daughter cells [6]. The countless processes in which SIRTs are involved make them potential targets for the treatment of chronic-degenerative diseases, including neurodegenerative disorders, cardiovascular diseases and cancer [7]. As cancer primarily comes from specific DNA alterations, the possibility of intervening more specifically compared to chemotherapy and radiotherapy, is being paid great attention [8].

Colorectal cancer (CRC) is the third most diagnosed and the second deadliest cancer worldwide, and its incidence is estimated to rise year by year [9]. Compared to the past, when CRC was mainly a disease of the elderly, its incidence is rising also in young adults, acquiring even more social relevance [10]. In this context, delineating the critical points of CRC carcinogenesis becomes pivotal. In recent years, the role of metabolic alterations in the development of CRC has emerged fiercely. Indeed, CRC is characterized by enhanced glycolysis and mostly repressed mitochondrial respiration and dynamics [11]. The role of important metabolic regulatory proteins including the peroxisome proliferator activated receptor (PPAR) family has been investigated, as PPARα activity in CRC mediates the oncogenic activity of mitochondrial 3-hydroxy-3-methylglutaryl-CoA synthase, thus promoting the expression of Src [12]. PPARγ regulates metabolism at different levels and its downregulation, determined by microRNA (miRNA) activity or promoter hypermethylation, can be found in CRC cells [13]. MiRNAs are single-stranded molecules playing a crucial role in regulating gene expression. Several studies showed that miRNAs can be used as biomarkers in diagnostic, prognostic and therapeutic fields in CRC [14]. In this regard, miR-21 can be considered as an important biomarker for CRC early diagnosis and a negative predictive factor for therapeutic response to 5-fluorouracil (5-FU), while miR-31 and miR-100 expression levels are positively correlated to Cetuximab resistance [14]. This evidence highlights how the development of a miRNA signature can provide a personalized therapeutic approach for CRC. ATP citrate lyase and sterol-regulatory element binding protein 1 (SREBP1) are pivotal factors in lipid metabolism. Their knockdown in vitro has been associated to poorer CRC survival, decreased aggressiveness and enhanced apoptosis [15]. Oncogenic protein Kirsten rat sarcoma (KRAS) is one of the most frequently mutated proteins in cancer and its function in CRC is related to glycolysis and pentose phosphate pathway promotion as well as lipid and amino acids metabolism enhancement [16]. Similar effects on CRC metabolism derived from the enzymatic activity of cystathionine β-synthase, c-Myc and Wnt [16]. Mitochondrial SIRTs SIRT2, SIRT3, SIRT4 and SIRT5 play a key role as metabolic regulators, finely operating on different cellular metabolic targets involved in cellular stress response and in energetic homeostasis. Within this framework, this review aims to give the readers an update on the involvement of mitochondrial SIRT2, SIRT3, SIRT4 and SIRT5 in chronic degenerative diseases focusing attention on their role in the development and progression of CRC.

## 2. Sirtuins as Modulators of Chronic Degenerative Diseases

Chronic degenerative diseases, such as diabetes, cardiovascular disorders, neurodegenerative diseases and cancer, represent the result of physio-pathological processes developing during life, strongly related to life habits including tobacco use, physical activity, nutrition and having the inflammaging state as a common thread [17,18,19]. Chronic inflammation and oxidative stress are finely controlled by the activation of nuclear factor κB (NF-κB), the signal transducer and activator of transcription 3 (STAT3), the mitogen-activated protein kinase (MAPK) and cyclooxygenase 2 (COX2) pathways [20,21]. SIRTs, interacting within these pathways, play a key role as regulators of these pathological processes [22,23].

Inflammaging represents a chronic inflammation state related to aging from which derives an accumulation of reactive oxygen species (ROS) which leads to cell damage, involved in the pathogenesis of multiple diseases frequently diagnosed in the elderly [17]. The anti-aging role of SIRT1 depends on its ability to counteract endothelial dysfunction, through the inhibition of NF-κB, and to promote autophagy by preventing p53 and mammalian target of rapamycin (mTOR) expression along with AMP-activated protein kinase activation [24,25]. The pro-longevity ability of SIRT1 can also be ascribed to its capacity to reduce the transcriptional activity of peroxisome proliferator-activated receptor-gamma coactivator-1α (PGC-1α) and to boost forkhead box O (FoxOs) function [24,25]. The relationship of SIRT2 with cellular aging has been showed in post-maturation bovine oocytes. SIRT2 gradually decreases during aging and its inhibition regulates autophagy and apoptosis [26]. SIRT2 is also able to prevent inflammaging development and insulin resistance by deacetylating NLR family pyrin domain containing 3 (NLRP3) in macrophages [27]. Multiple SIRT6 pathways are related to senescence, oxidative stress and inflammation. SIRT6 counters oxidative stress by inhibiting c-Jun, Notch1 and Notch4 transcription and protein kinase B (AKT) phosphorylation via metal regulatory transcription factor 1 (MTF1) and BRG1-associated factor 170 (BAF-170) recruitment, resulting in the downregulation of inflammatory cytokines expression [28]. SIRT7 regulates stem cell senescence and heterochromatin stabilization; its deficiency accelerates the aging process while its activity displays a beneficial role [29]. 

Oxidative stress and chronic inflammation represent important susceptibility factors for the development of metabolic diseases such as diabetes, thus worsening the prognosis and the pathological burden [30]. Accumulating evidence indicates that SIRT1 and SIRT6 are crucial players in chronic inflammation related to vascular homeostasis, cardiovascular disease, cardiac dysfunction and diabetes [31]. In obesity and prediabetes, SIRT6 expression in visceral adipose tissue seems to be lower than in normal condition, while there is higher expression of pro-inflammatory factors, such as NF-kB suggesting a role for this SIRT as a regulator of inflammation in this pathological condition [32]. In the same clinical context, a lower SIRT1 expression levels have been observed at a subcutaneous fat level [33]. The downregulated SIRT1 expression is correlated with pro-inflammatory cytokines production and reduced myocardial function [33]. Diabetic hyperglycemia occurs via methylation on SIRT6 promoter, thus determining SIRT6 downregulation in TeloHAEC cells exposed to high glucose concentrations [34]. In vitro exposure to high glucose concentrations also determined downregulation of SIRT6 in podocytes, causing mitochondrial alteration, increased ROS production and phosphorylation of AMP-activated protein kinase (p-AMPK) [35]. Again, in cardiac microvascular endothelial cells treated with high glucose and palmitic acid, the downregulation of SIRT6 expression has been provided. Additionally, the in vitro knockout of SIRT6 promoted the endothelial-to-mesenchymal transition and aggravated the diabetic cardiomyopathy through Notch1 signaling pathway activation [36]. In diabetic and gestational diabetic patients SIRT1 downregulation, along with the downregulation of FKBP prolyl isomerase-like (FKBPL), has been linked to vascular dysfunctions and pregnancy complications both in vivo and in vitro [37]. The development of diabetic nephropathy is also affected by SIRT1 expression levels. The upregulation of this SIRT displays renoprotective effects by intervening in numerous cellular pathways, from metabolism to cell survival [38]. Moreover, SIRT1 mediates the effects of metformin via AMPK/SIRT1-FoxO1 pathway, reduces oxidative stress and promotes autophagy, thus countering diabetic kidney pathological process [39]. Mitochondrial SIRT3, SIRT4 and SIRT5 also participate in the pathogenesis of diabetes, through the modulation of glucose uptake and tolerance, control of insulin secretion and sensitivity and fatty acid metabolism [40]. In type 2 diabetic patients, SIRT3 downregulation has been found in pancreatic islets cells, implying a role in regular cell function and insulin secretion [40]. In diabetic and prediabetic conditions, the activated inflammatory pathways cause an epigenetic upregulation of fatty acid-binding protein 4 (FABP4) which, in turn, determines decreased expression of SIRT3 in macrophages and abnormal tissue repair [41]. 

Chronic inflammation strongly relates to atherosclerosis by promoting plaque formation and instability [42]. It has been shown that the activation of inflammatory cascade in diabetic atherosclerotic plaques, associated with lower SIRT6 levels, induce the expansion of oxidative and inflammatory processes in endothelial cells [43]. On the contrary, SIRT6 overexpression in macrophages promoted plaques stability through autophagic flux activation, apoptosis inhibition, reduced in plaque infiltration and cell adhesion molecules expression, such as vascular cell adhesion molecule-1 (VCAM-1), intercellular adhesion molecule-1 (ICAM-1), and platelet selectin (P-selectin) [44]. SIRT1 involvement in the atherosclerotic process is related to its ability in mediating the estrogen signaling cascade, thus protecting arteries functionality [45]. SIRT1 expression is also promoted by laminar flux, which determines deacetylation and repression of yes-associated protein 1 (YAP), inhibition of the pro-inflammatory gene expression and at the same time promoting autophagy and protecting from atheroma development [46]. 

Vascular homeostasis, endothelial functionality and blood pressure are associated to ROS levels and inflammatory status, which determine reduction in nitric oxide (NO) bioavailability provoking the absence of NO-dependent vasodilatation [47]. SIRT3 functionality is involved in the pathogenesis of hypertension. Indeed, its downregulation in endothelial cells is essential in determining the vascular alterations and dysfunction, proper of hypertension [48]. At the same time, SIRT6 activity prevents hypertension and its renal complications by maintaining endothelial homeostasis through the induction of GATA-binding protein 5 (GATA5) via NK3 homeobox 2 (Nkx3.2) transcription inhibition [49]. 

Due to the ability to modulate aging-associated modifications, SIRTs are also under the spotlights in the neurodegenerative context, such as Parkinson and Alzheimer diseases [50]. In the central nervous system, SIRT1 and SIRT3 display neuroprotective role, while SIRT2 activity is associated to neurodegeneration [50]. Intriguingly, these SIRTs also affect enteric nervous system functionality by modulating different hormones expression [51]. The loss of SIRT3 can accelerate the development of neurodegenerative diseases, while its overexpression can prevent them. This property achieves SIRT3 as a relevant target of interest for the comprehension of neurodegeneration and inspiration for new target therapies [52]. A recent study described the stigmasterol treatment in the prevention of neurodegeneration mechanisms, elucidating the neuroprotective role of SIRT1 against hydrogen-peroxide-induced toxicity [53]. In vitro studies on human neuronal cells showed that stigmasterol activates SIRT1 which, in turn, deacetylates and activates catalases, thus promoting the production of antioxidants and the resistance against oxidative stress [53]. 

The skeletal system is commonly involved in the pathological context of the elderly. In this scenario, SIRT1 deacetylase activity has been shown to be important in the development of osteoporosis and osteoarthritis through the deacetylation of forkhead box proteins Os (FoxOs) promoting bone formation and countering bone resorption in skeletal cells [54]. SIRT1, SIRT2, SIRT3 and SIRT6 have been related to intervertebral disc degeneration, one of the main causes of age-related low back pain. Preclinical studies have reported that sirtuin activators such as resveratrol, showed beneficial effects against inter-vertebral disc degeneration [55]. 

Due to pivotal role in the modulation of multiple pathways involved in the onset and development of chronic and age-related disorders having inflammaging as a common factor, sirtuin are attractive targets in the pathophysiology of many conditions, including cancer.

## 3. Sirtuins and Cancer

Aging, inflammation, cellular stress and ROS accumulation are important players in cancer initiation and progression [56], triggering a series of changes in cellular genetics and epigenetics from which results alterations in cellular phenotype and metabolism. In this scenario, SIRTs act as tumor promoters and/or tumor suppressors, depending on cancer type [2]. The pleiotropic roles of SIRT in many metabolic pathways such as tricarboxylic acid cycle (TCA), lipid metabolism and even urea cycle, strongly linked to progression and tumor survival, make these proteins as main character in the most widespread cancers [57].

Breast cancer is one of the most frequently diagnosed cancer worldwide. It is the most common cancer in the USA and a frequent cause of death, particularly in women, with its incidence rising in China [58]. In breast cancer, stabilization of mitochondrial glutaminase (GLS) by SIRT5-mediated desuccinylation leads to enhanced proliferation and survival, serving substrates for numerous metabolic pathways, correlating to poorer breast cancer prognosis when found overexpressed [59]. SIRT7 expression also affects breast cancer progression and metastasis deacetylating SMAD4 and, consequentially, determining its degradation through β-transducin repeat-containing protein (β-TrCP1), while its deficiency is linked to transforming growth factor β (TGF-β) pathway activation enhancing metastatic process [60]. In MCF-7 breast cancer cells, SIRT7 inhibition promotes stress resistance, downregulates insulin receptor (INSR), and modulates insulin-like growth factors (IGFs) pathways affecting cell metabolism and growth [61]. SIRT4, able to inhibit cancer stem cells proliferation and survival, is found downregulated in breast cancer, and its levels are inversely correlated to SIRT1 expression and linked to acquisition of breast cancer aggressive phenotype [62].

In prostate cancer, the most diagnosed male cancer in industrialized society, where different environmental factors concur to its high incidence [63], SIRT4 affirms its role as tumor suppressor. SIRT4 is involved in p21-activated kinase 6 (PAK6) pathway, driving adenine nucleotide translocator 2 (ANT2) deacetylation and promoting apoptosis [63]. In detail, SIRT4 contrasts PAK6 cell death inhibition as PAK6 is meant to induce SIRT4 proteolysis and ANT2 phosphorylation [64]. SIRT5 affects prostate cancer proliferation, progression and survival enhancing Acetyl-CoA acetyltransferase 1 activity leading to MAPK pathway promotion and ultimately to Matrix metallopeptidase 9 (MMP-9) and cyclin D1 expression [65]. An important role in cell cycle is carried out by Cyclin-dependent kinase 12 (CDK12), which can be found mutated in a subgroup of prostate cancer patients with infiltration of CD4+forkhead boxP3 (FOXP3)- T-cells and neoantigens exposition, thus resulting in an exalted immunotherapy response [66]. Therefore, CDK12-mutated cancers might constitute a separate subgroup of prostate cancer in which immunotherapy may be effective [66]. SIRT7 activity displays a crucial role in prostate cancer where its depletion in vitro and in vivo causes inhibition of cell proliferation, autophagy and invasion via the downregulation of androgen receptor (AR) pathway mediated by small mother against the decapentaplegic 4 (SMAD4) protein [67]. AR pathway activation also leads to SIRT3 transcriptional repression, thus enhancing mitochondrial aconitase activity and driving prostate cancer malignancy and bone invasion [68]. In contrast, SIRT3, along with SIRT6, can also be considered as hallmarks of poorer prognosis in prostate cancer patients as these SIRTs inhibit necroptosis receptor-interacting protein kinase 3 (RIP3/RIPK3) pathway and immune response, while when knocked down in vitro immune response and necroptosis recovery occurred [69]. SIRT1 promotes EMT in prostate cancer upregulating Snail and Twist or inhibiting of the expression of E-cadherin [70].

Pancreatic cancer, the deadliest and malignant cancers as a consequence of the difficulty to identify it in early stages [71], is also influenced by SIRTs activity and expression. SIRT5 expression is directly correlated to positive prognosis, as its loss promotes glutamic-oxaloacetic transaminase 1 (GOT1) acetylation, thus promoting cell proliferation by enhancing glutamine and glutathione metabolism [72]. In vitro and in vivo upregulation of SIRT6 by tumor suppressor Krüppel-like factor 10 (KLF10) activity affects glycolysis, epithelial–mesenchymal transition and metastasis [73]. 

Oral cancer represents a relatively non frequent disease, largely preventable but burdened by a high mortality rate [74]. SIRT7 antagonizes oral cancer metastasis through the deacetylation of small mother against decapentaplegic 4 (Smad4) and can be found frequently downregulated in vivo because of miR-770 expression, whose expression represents a negative prognostic factor [75]. In oral squamous carcinoma cells Cal27, SIRT1 determines cell apoptosis affecting procaspase-3 and cyclin B1, ultimately exercising an anticancer effect [76]; additively, SIRT1 has been demonstrated to induce E-cadherin expression and to repress transforming growth factor-beta (TGF-β) interfering with the acquisition of the metastatic phenotype [77].

CRC represents the most common diagnosed cancer worldwide with its incidence rising year by year [10]. Recent evidence highlighted the antitumorigenic action of SIRT6 in CRC. SIRT6 indirectly opposes the intracellular lipid droplets (LDs) formation. Indeed, while LDs sustain human CRC cell metabolism and growth, SIRT6, activated by the FOXO3 transcription factor, counteracts their storage impairing lipid synthesis. The increase in LDs density depends on the cellular signal derived from the binding of epidermal growth factor (EGF) to its receptor EGFR that causes the inhibition of FOXO3/SIRT6 axis and, consequently, leads to a strong increase in LDs production [78]. In CRC cells, SIRT6 expression promotes autophagic and apoptotic effect [79], while SIRT1 overexpression induced by interleukyne-1β is associated to increased malignancy [80]. SIRT1 shows an astonishing duality in cancer. In CRC, SIRT1 acts as an oncogene and can be found upregulated in conditions of depletion of SMAD-specific E3 ubiquitin protein ligase 2 (SMURF2) [81]. The upregulation of SIRT1 by for Forkhead Box Q1 (FOXQ1) contributes to CRC survival inducing cancer cell radio-resistance and cell stemness [82]. SIRT1, consequently to its phosphorylation and stabilization through c-Jun N-terminal kinases (JNK), favors CRC development in HCT116 cells through the deacetylation of the Snail protein and the subsequent upregulation of the expression of interleukin-6 and interleukin-8 [83]. On the other hand, SIRT1 can suppress tumor initiation, increasing genome stability and inhibiting inflammation at the pre-cancer stage. Overexpression of SIRT1 can induce cell cycle arrest via E2F Transcription Factor 1 (E2F1) and represses CRC proliferation and tumor initiation [70]. SIRTs play a major role in cancer through the modulation of several autophagy-related pathways, thus eliminating unnecessary or dysfunctional intracellular components. Particularly, SIRT2 promotes human CRC cells growth, deacetylating FOXO1 which is impaired in its binding to autophagy-related gene 7(ATG7), countering autophagy-induced apoptosis [84]. Among cancerous cells, CRC cells manifest altered oxidative metabolism, impairing oxidative phosphorylation in favor of aerobic glycolysis, the Warburg’s effect, gaining a great number of substrates able to promote cell survival and proliferation [85,86]. SIRT1 and SIRT6 regulate the Warburg effect by modulating glycolysis associated genes expression such as glucose transporter type 1 (GLUT1) and glyceraldehyde-3-phosphatede-hydrogenase (GAPDH) [87]. c-Myc, one of the most important oncogenes, boosts glutamine metabolism and provides a wide variety of energetic substrates and nucleotides to cancerous cells [88]. The center of these key metabolic pathways is the mitochondrion, which plays a critical role from energy generation to metabolism and ion homeostasis. Subtle changes in metabolic processes, such as a switch from oxidative phosphorylation (OXPHOS) to glycolysis, can alter the phenotype of normal to cancerous cells. In the earliest stages of tumor development, non-oxidative metabolism promotion, subsequent to mitochondrial pyruvate carrier (MPC) downregulation, can be considered a signature of CRC initiation [89]. Metabolic reprogramming allows cancer cells to adapt to the tissue microenvironment and boosts cancer progression. CRC cells display upregulation of many mitochondrial enzymes, such as tumor necrosis factor type 1 receptor-associated protein (TRAP1), which inhibits OXPHOS via downregulation of cytochrome oxidase (the complex IV of the respiratory chain) and SDH (succinate dehydrogenase), NF-κB, involved in the regulation of mitochondria dynamics, respiration, gene expression and metabolism, OMA1 zinc metallopeptidase (OMA1) mitochondrial protease, which promotes metabolic reprogramming and further CRC development; heath shock protein 60 (HSP60) mitochondrial chaperone, responsible for maintaining mitochondria proteostasis, PGC-1α, which controls energy metabolism, mitochondrial biogenesis and homeostasis and several metabolic pathways responsible for drug resistance; and the Hippo–Yap pathway involved in development, growth, repair, and homeostasis [90]. Many studies assert that mitochondrial SIRTs are responsible for various post-translational modifications, such as deacetylation, decrotonylation, desuccynilation and demalonylation and control the functionality of numerous proteins involved in metabolism. The knowledge of their action and interplay in modulating metabolic and energetic homeostasis is particularly fascinating as it might help in designing new prevention/therapeutic strategies against CRC. 

## 4. Mitochondrial Sirtuins and CRC

### 4.1. SIRT2 in CRC Metabolism

SIRT2, firstly described at cytosolic and nuclear level, has been also found in the mitochondria [91]. Particularly, SIRT2 localizes in mitochondria intima and seems to be important in maintaining mitochondrial homeostasis. To date, the role of SIRT2 in CRC is still controversial. Recently, SIRT2 has been shown to operate as a deacetylase targeting isocitrate dehydrogenase 1 (IDH1) at lysine 224 enhancing its enzymatic activity and, ultimately, inhibiting hypoxia inducible factor α (HIF1α)/Src pathway, exerting a tumor suppressor activity in CRC cells (Figure 1) [92]. SIRT2 activity also inhibits proliferation, as it has been observed that this enzyme is a target of the Wnt/β-catenin pathway [86]. In fact, SIRT2 overexpression, as well as Wnt/β-catenin pathway inhibition, determines increased OXPHOS levels and CRC differentiation (Figure 1) [93]. On the contrary, many other studies demonstrate an oncogenic activity for SIRT2. This SIRT interacts directly with signal transducer and activator of transcription 3 (STAT3) regulating its phosphorylation and its translocation in the nucleus, thus causing an increase in vascular endothelial growth factor A (VEGFA) secretion furthering angiogenesis, tumor survival and growth (Figure 1) [94]. SIRT2 inhibition through its specific inhibitor AK-1 determines degradation of Snail transcription factor subsequently upregulating p21 and provoking cell cycle arrest and slow proliferation in HCT116 and HT29 cells [95]. In CRC, it has been reported that lysine crotonylation of α-enolase (ENO1) favors tumor development and metastasis. In this regard, SIRT2 can exert decrotonylation activity countering crotonylation of ENO1 mediated by CREB-binding protein (CBP) interfering in this pro-tumoral pathway (Figure 1) [96]. SIRT2 importance in colorectal cancer is also assessed by the evidence that a SNP (small nucleotide polymorphism) in 3′-UTR (untranslated) of SIRT2 sequence impeding hsa-miR-376a-5p binding to SIRT2 can increase patient susceptibility to colorectal cancer development [97].

### 4.2. SIRT3 and CRC Metabolism

SIRT3, the most extensively studied mitochondrial SIRT, is involved in various metabolic processes related to metabolism and mitochondrial homeostasis. Several studies have shown the association of SIRT3 with different human diseases, including age-related diseases, cancer, heart and metabolic diseases [98]. Mitochondrial homeostasis is maintained thanks to endogenous regulators that modulate the expression and function of SIRT3 in response to different conditions of cellular stress. Mitochondrial biogenesis and metabolism are coordinated by many genes such as PGC1α, nuclear respiratory factor 1 (NRF1), transcription factor A, mitochondrial (TFAM), cytochrome c oxidase (complex IV) (COX IV), mitochondrial uncoupling protein 2 (UCP2), mitochondrial uncoupling protein 5(UCP5) (Figure 2). SIRT3s influence the expression of these genes impairing mitochondrial dynamics which determine a reduction in oxygen consumption and affect malignant properties of CRC cells [99]. SIRT3 regulates mitochondrial DNA repairing cellular skills, exerting a role in controlling the cancerous transformation, associated with the accumulation of DNA damages [100,101,102,103]. SIRT3 deacetylates proteins for whose function the adequate acetylation level is required, such as mutY DNA glycosylase (MUTYH), nei-like DNA glycosylase 1 (NEIL1), nei-like DNA glycosylase 2 (NEIL2), 8-oxoguanine DNA glycosylase (OGG1), human apurinic/apyrimidinic endonuclease (APE1) and DNA ligase 3 (LIG3) [104,105]. The one carbon metabolism role in CRC tumorigenesis has been extensively studied [106,107,108,109]. SIRT3 affects one carbon metabolism by promoting the activity of serine hydroxy methyltransferase 2 (SHMT2), whose overexpression is linked to poorer prognosis for CRC patients, thus triggering cell proliferation and growth [110]. SIRT3 is also the major deacetylase of methylenetetrahydrofolate dehydrogenase 2 (MTHFD2), promoting its activity and determining a boost in cell proliferation and cancer development [111]. The activity of SIRT3 relates to the metabolism of nitric oxide (NO), with a controversial role in CRC [112]. The mitochondrial nitric oxide synthase 1 (NOS1) promotes the activity of SIRT3 which, in turn, mediates the NOS1-related apoptotic resistance effect by modulating the levels of intracellular ROS and reducing the acetylation levels of SOD2 [113]. In CRC, SIRT3 activates manganese superoxide dismutase (MnSOD) [114], thus influencing the non-tumor adjacent tissue which displays high levels of proteins able to detoxify hydrogen peroxide [115]. The controversial role of SIRT3 is also confirmed by its involvement in the regulation of gut homeostasis and mucosal barrier function which influence tumorigenesis and determine the appearance of alterations in gut microbiome [116]. SIRT3-deficient mice showed hyper-susceptibility to colon inflammation and CRC development as a consequence of alteration of intestinal integrity, abundance of microbial pathogens and p38 signaling promotion [116]. The chemotherapeutic involvement of SIRT3 in CRC is demonstrated by its interaction with anticancer drugs, influencing the success of chemotherapy. Indeed, SIRT3 is downregulated by cisplatin with consequent hyperacetylation and inhibition of MTHFD2 [111]. On the contrary, SIRT3 overexpression led to hyperactivity of MTHFD2 and poorer prognosis, underlining SIRT3 as possible prognostic biomarker for CRC [111]. Conversely, SIRT3 ability to decrease ROS levels interferes with oxaliplatin-induced apoptosis, decreasing the chemotherapy effect [113]. SIRT3 levels represent predictive factor of chemotherapeutic success increasing PGC1α and SOD2 expression, consequentially determining reduction in mitochondrial ROS (mtROS) and increased drug resistance (Figure 2) [117]. Emerging prevention strategies to counteract the development of CRC also include nutrition interventions, given the ability of many nutrients to act as exert anticancer effects through the modulation of SIRTs. In this context, mitophagy can be induced via modulation of SIRT3 by δ-valerobetaine, a newly discovered epi-nutrient occurring ruminant milk [118]. SIRT3 induces mitophagy and apoptotic cell death through PTEN-induced kinase 1(PINK1)/Parkin/microtubule-associated proteins 1A/1B light chain 3B (LC3B) axis in CRC cells (Figure 2) [118]. In vitro studies revealed that the ability of on milk whey to affect HT-29, HCT 116, cell proliferation, cell cycle, apoptosis and metabolism via upregulation of SIRT3 and PPAR along with downregulation of lactate dehydrogenase A (LDHA), SREBP1 and PPARα (Figure 2) [119]. Intriguingly, another food-derived betaine, ergothioneine, exerts anti-CRC effect through SIRT3 activation [120]. In vitro ergothioneine treatment can induce cell death via ROS accumulation and necroptotic mixed lineage kinase domain like pseudokinase (MLKL) pathway activation promoting SIRT3 activity and its interaction with MLKL (Figure 2) [120].

### 4.3. SIRT4 in CRC Metabolism and Prognosis

The activity of SIRT4 in CRC is still relatively unknown. However, many findings point out the tumor suppressor role of this mitochondrial SIRT able to counter the gaining of malignant properties from cancerous cells, underlining that SIRT4 expression is lower in CRC compared to normal tissue [121]. SIRT4 activity affects glutamine metabolism repressing the activity of glutamate dehydrogenase (GDH) and abrogating alpha-ketoglutarate (αKG) formation (Figure 3) [122]. The SIRT4-mediated upregulation of E-cadherin and downregulation of proteins associated to cellular mobility and metastasis make this protein a valuable prognostic biomarker in CRC patients [123]. Indeed, low SIRT4 expression is related to worse prognosis and aggressiveness of CRC [124]. SIRT4 expression also influences the response of CRC to chemotherapy. It has been observed that SIRT4 activity can be determinant in defining the apoptosis induced by 5-fluorouracil (5-FU), one of the most used drugs in colorectal cancer therapies [125,126]. The suppression of SIRT4 promotes glutaminase (GLS) activity, thereby initiating the activation of protein kinase B (AKT). In CRC, SIRT4 inhibits proliferation, migration, and invasion regulating the AKT/glycogen synthase kinase 3β (GSK3β)/CyclinD1 pathway (Figure 3) [127].

### 4.4. SIRT5 and CRC Metabolism, Prognosis and Therapy

Unlike other SIRTs, SIRT5 does not exhibit a unique affinity for deacetylation but displays protein desuccinylation, demalonylation, and deglutarylation activity. In recent years, insights on the role of SIRT5 in CRC have established its participation in metabolic processes such as glucose oxidation, ketone body formation, fatty acid oxidation, ammonia detoxification and ROS production [128]. Firstly, the cell survival promoting activity of SIRT5 can be referred to its ability to promote the autophagic process [129], important for CRC development. SIRT5 promotes autophagy by deacetylating lactate dehydrogenase B (LDHB) and enhancing lysosome acidification (Figure 4) [130]. SIRT5 deglutarylation activity on glutamate dehydrogenase 1 (GLUD1) promotes CRC carcinogenesis by enhancing glutaminolysis and determining the formation of TCA intermediates [131]. Krebs Cycle promotion also occurs via SIRT5 desuccinylation of citrate synthase (CS) and enhancement of its enzymatic activity, boosting CRC proliferation and migration [132]. Likely to SIRT3, SIRT5 acts one carbon metabolism through desuccinylation of SHMT2 resulting in CRC enhanced cell survival and growth also in condition of deprivation of glycine and serine (Figure 4) [133]. SIRT5 expression can be considered a useful CRC prognostic biomarker by affecting chemotherapy in response to FOLFOX regimen (leucovorin, 5-FU and oxaliplatin) and cetuximab treatment (Figure 4) [134]. SIRT5 determines succinate accumulation through the demalonylation of succinate dehydrogenase complex subunit A(SDHA), promotes thioredoxin 2 (TrX2) activity, responsible of FOLFOX resistance, and increases succinate/αKG ratio, responsible of cetuximab resistance (Figure 4) [134]. SIRT5 interferes with FOXO3a-induced apoptosis reducing its acetylation levels. Recent evidence shows that SIRT5 modulators, suramin and resveratrol, reduce HCT116 cell viability, with more effect in p53^+/+^ compared to p53^−/−^ HCT116 cells [135].

## 5. Conclusions and Future Perspectives

In recent years, many efforts are being focused on finding novel therapeutic and prevention strategies to fight cancer. Many pathogenetic factors are still to be discovered and understood and therapies are still not free from serious collateral effects for the patients. In this scenario, without gene expression alterations and through reversible specific modifications, the epigenetics field seems to be a promising path. SIRTs are involved in many fundamental cellular pathways, controlling DNA histones acetylation and post-translational modifications [1]. The possibility of using SIRTs as biotargets in cancer prevention and therapy is intriguing. Many results support the efforts to find SIRT-targeting molecules for CRC treatment. Metformin showed cytotoxic property in HT-29 cells, increasing ROS levels and SIRT3 activity [136]. Evodiamine, a compound extracted from Tetradium plants, boosts SIRT1 activity, inhibits NF-κB pathway and suppresses the metastatic potential of HT-29 and HCT116 cells [137]. BZD9L1, a SIRT1/2 inhibitor, shows in vitro anti-cancer activities in HCT116 cells, subsequently to regulation of major cancer pathways and apoptosis induction through p53, one of the most important regulators of cell cycle and DNA repair [138]. P53 acts as a transcription factor with tumor suppressor function and has been found mutated in codon 72 in a great percentage of CRC samples, promoting the acquisition of malignant properties [139]. MHY2245, a derivative of sirtinol, decreases SIRT1 activity in o HCT116 cells and causes DNA damage, provoking cell cycle arrest and programmed cell death [140]. MHY2256 suppresses SIRT1/2 expression and promotes the acetylation of FOXO1, ultimately inhibiting cell growth in HCT116, HT-29 and DLD-1 cell lines [141]. Recent studies described UBSC039 as a new SIRT6 activator. In several CRC cell lines, the activation of SIRT6 leads to enhanced ROS production, followed by an increased autophagy mediated by Beclin1 and autophagy-related gene5 (ATG5) [142]. MDL-811 activates SIRT6 in vitro and in vivo with antiproliferative effects [143]. Moreover, MDL-811 targets cytochrome P450 family 24 subfamily A member 1 (CYP24A1) consequentially promoting the anti-CRC effect of vitamin D [144]. 

It should not be overlooked that prevention strategies against CRC are based on dietary interventions, as a strong link has been observed between the risk of CRC and diet-related inflammation [145]. On the other hand, a healthy and balanced diet interferes with CRC development through downregulation of inflammatory pathways, neutralization of free radicals, regulation of normal intestinal integrity, promotion of immune response and modulation of microbiota composition [146,147,148,149]. Nutritional components can intervene in CRC pathogenesis even acting epigenetically. Nutrients acting as *epi-nutrients* show the ability to activate SIRTs [76,118,119,120]. Among *epi-nutrients*, ergothioneine and δ-valerobetaine modulate SIRT3 and SIRT6 and oppose cancer viability and progression [76,118,119,120]. In addition, mangiferin and resveratrol show in vitro anticancer properties in CRC via SIRT/MGF and SIRT1/NF-κB signaling, respectively [135,150,151]. These are just some of the effects of epi-nutrients, extensively described in recent excellent reviews [152,153,154], showing the potential of mitochondrial SIRTs in the control of the development and progression of CRC. In this regard, a balanced diet, that can guarantee the maintenance of optimal epi-nutrient levels, can represent an important starting point both for primary prevention strategies and for the success of chemotherapy. In CRC, mitochondrial SIRTs, acting as metabolic regulators, represent promising prevention and therapeutic targets (Figure 5) in the perspective of a precision medicine, which starts from a personalized nutrition for age and disease.

## Figures and Tables

**Figure 1 ijms-23-03212-f001:**
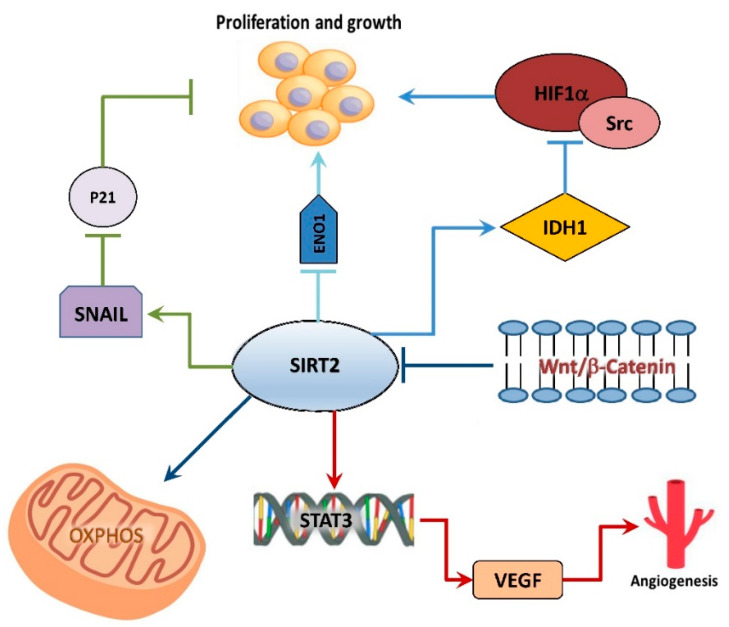
SIRT2 acts both as tumor suppressor and tumor promoter, inhibiting tumor proliferation but also promoting tumor survival and the angiogenetic process. HIF1α, Hypoxia inducible factor 1α; IDH1, Isocitrate dehydrogenase 1; STAT3, signal transducer and activator of transcription 3; VEGFA, vascular endothelial growth factor A; ENO1, α-enolase.

**Figure 2 ijms-23-03212-f002:**
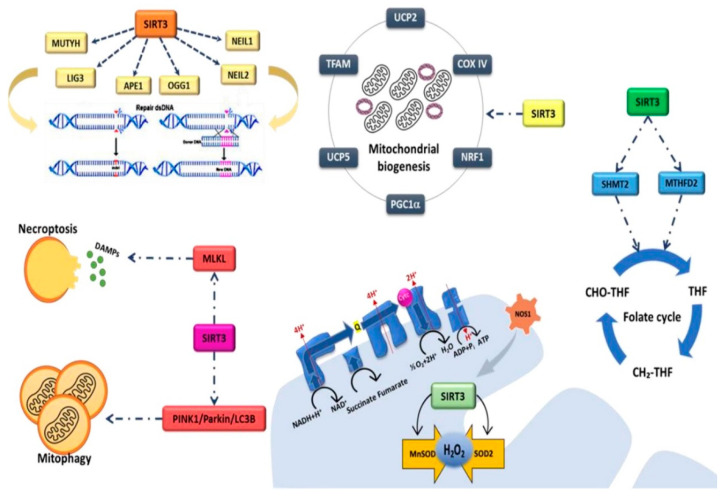
SIRT3 pathways in CRC. SIRT3 activation regulates mitochondrial homeostasis and promotes metabolic alterations deeply influencing CRC cells phenotype. PGC1α, peroxisome proliferator-activated receptor alpha; NRF1, nuclear respiratory factor 1; TFAM, Transcription factor A, mitochondrial; UCP2, Mitochondrial uncoupling protein 2; UCP5, Mitochondrial uncoupling protein 5; COX IV, Cytochrome c oxidase (complex IV); MUTYH, mutY DNA glycosylase; LIG3, DNA Ligase 3; APE1, Human apurinic/apyrimidinic endonuclease; OGG1, 8-Oxoguanine DNA Glycosylase; NEIL1/2, Nei Like DNA Glycosylase 1/2; MLKL, Mixed Lineage Kinase Domain Like Pseudokinase; PINK1, PTEN-induced kinase 1; LC3B, microtubule-associated proteins 1A/1B light chain 3B; MnSOD, Manganese superoxide dismutase; SOD2, superoxide dismutase 2; NOS1, nitric oxide synthase 1; SHMT2, Serine Hydroxy methyltransferase 2; MTHFD2, Methylenetetra-hydrofolate Dehydrogenase 2.

**Figure 3 ijms-23-03212-f003:**
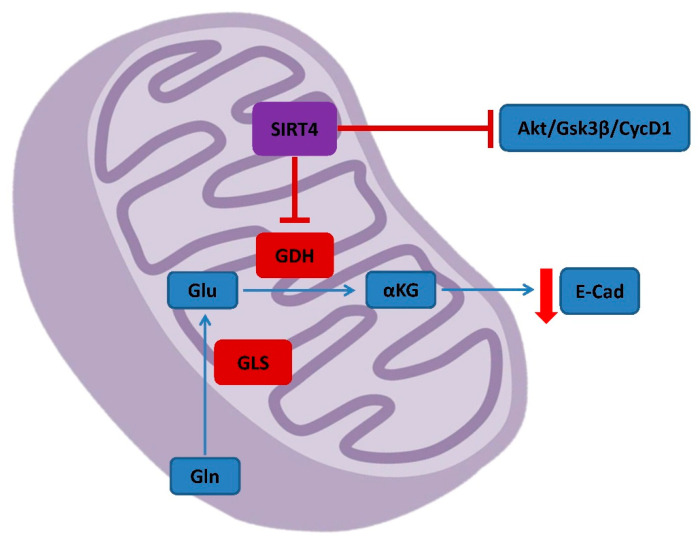
Role of SIRT4 in CRC. SIRT4 is mainly suppressive and counteract the activation of proliferation and the accumulation of important metabolites originated from glutamine metabolism. Akt, Protein kinase B; Gsk3β, Glycogen synthase kinase 3β; CycD1, CyclinD1; GDH, Glutamate dehydrogenase; Glu, glutamate; αKG, α-ketoglutarate; E-Cad, E-Cadherin; GLS, glutaminase; Gln, glutamine.

**Figure 4 ijms-23-03212-f004:**
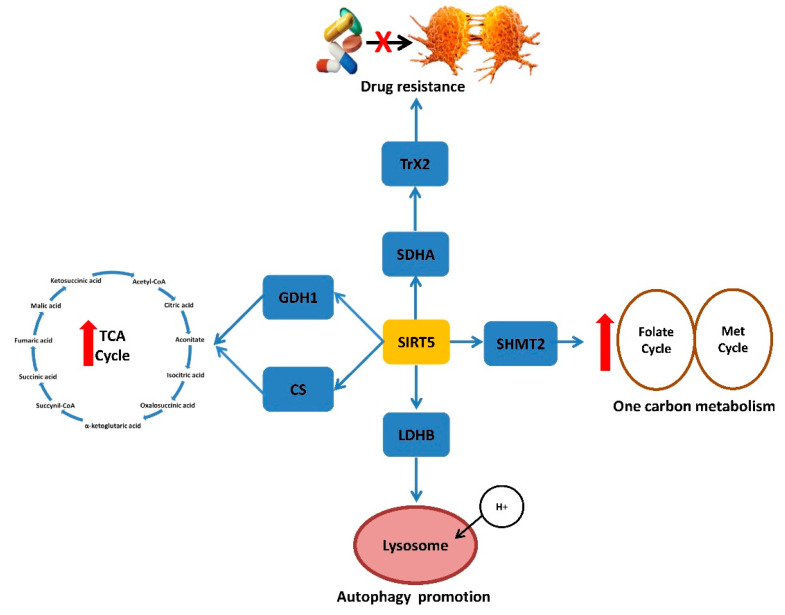
SIRT5 activity confers proliferative advantages to CRC cells enhancing energetic metabolism, cell renewal and chemoresistance. TCA, tricarboxylic acid cycle; GDH1, glutamate dehydrogenase 1; CS, citrate synthase; TrX2, thioredoxin 2; SDHA, succinate dehydrogenase complex subunit A; SHMT2, serine hydroxy methyltransferase 2; LDHB, lactate dehydrogenase B.

**Figure 5 ijms-23-03212-f005:**
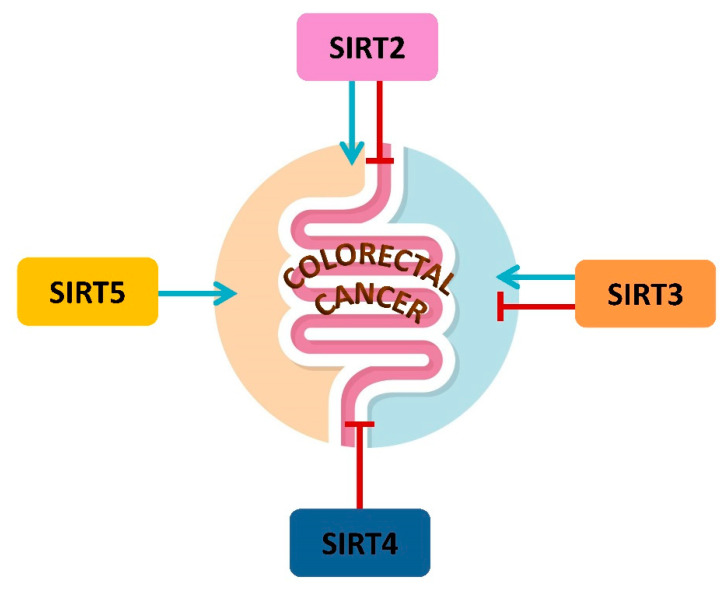
Mitochondrial SIRTs, SIRT2, SIRT3, SIRT4 and SIRT5, exert different effects on CRC initiation, promotion and progression. The blue arrows stand for an oncogenic activity, while the red ones indicate a tumor suppressive role. SIRT2 and SIRT3 exert a multifaceted activity on CRC, SIRT4 is mainly a tumor suppressor while SIRT5 is mainly a tumor promoter.

## Data Availability

Not applicable.

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
