# Peer review of "Mitochondrial Sirtuins in Chronic Degenerative Diseases: New Metabolic Targets in Colorectal Cancer"

_ijms, 2022, doi:10.3390/ijms23063212_

Round 1

Reviewer 1 Report

In this article, the authors give an update on the involvement of mitochondrial sirtuin SIRT2, SIRT3, SIRT4 and SIRT5 in chronic-degenerative disease focusing attention on their role in the development and progression of CRC. The manuscript is straightforward, well written, and concise and has clear results, within the scope of a review article. Definitely deserves to be published and is a valuable contribution to the “International Journal of Molecular Sciences” journal. Some minor comments need to be addressed before publication.

Minor points:

[1]1. Introduction”, Page 2 of 23, Lines 72-92:

PPARÉ£ regulates metabolism at different levels and it has been observed that its reduced activity determined by microRNAs (miRNAs) activity or promoter hypermethylation can be found in CRC [14].”.

At that point the authors should highlight the potential utility of miRNAs as biomarkers in either tissues or blood for the assessment of response to the agents implemented in CRC, including the 5-fluorouracil based therapies, and EGFR inhibitors. They regulate critical pathways involved in the CRC pathogenesis, including the p53, PI3K, RAS, MAPK, EMT transcription factors, and Wnt/β-catenin pathways. Development of miRNA signature for predicting treatment response designates a personalized therapeutic approach of CRC.

Recommended reference: Boussios S, et al. The Developing Story of Predictive Biomarkers in Colorectal Cancer. J Pers Med. 2019;9(1):12.

[2] “3. Sirtuins and cancer”, Page 5 of 23, Lines 224-227:

SIRT5 affects prostate cancer proliferation, progression and survival enhancing acetyl-CoA acetyltransferase 1 activity leading to MAPK pathway promotion and ultimately to Matrix metallopeptidase 9 (MMP-9) and cyclinD1 expression [64].”.

Cyclin D1 plays an important role in cell cycle progression through the association with CDK12. This is important as it has been reported that CDK12 mutations may delineate an immuno-responsive subgroup of prostate cancer with increased levels of T-cell infiltration and neoantigens. Based on that, CDK12-mutated tumors might constitute a separate subgroup of prostate cancer in which immunotherapy may be effective.

Recommended reference: Ghose A, et al. Genetic Aberrations of DNA Repair Pathways in Prostate Cancer: Translation to the Clinic. Int J Mol Sci. 2021;22(18):9783.

[3] “5. Conclusions and future perspectives”, Pages 11 and 12 of 23, Lines 466-468:

BZD9L1, a SIRT1/2 inhibitor, shows in vitro anti-cancer activities in HCT116 cells, subsequently to regulation of major cancer pathways and apoptosis induction through p53 [131].”.

This should be highlighted as TP53 protein, encoded by the TP53 tumor suppressor gene, has a regulatory role in cell growth arrest, DNA repair and apoptosis but also in oxidative stress, DNA damage and cell aberrant proliferation, thus maintaining cell cycle homeostasis. Mutated TP53 is expressed in about 60% of CRC. Point mutation in codon 72 - resulting in the substitution of proline to arginine - leads to dysfunction of the cell cycle “gatekeeper” that promotes the malignant process.

Recommended reference: Zarkavelis G, et al. Current and future biomarkers in colorectal cancer. Ann Gastroenterol. 2017;30(6):613-621.

Author Response

Reply to Reviewer comments

Manuscript ID: ijms-1634624

Mitochondrial sirtuins in chronic degenerative diseases: new metabolic targets in colorectal cancer

Reviewer1

Comments and Suggestions for Authors

In this article, the authors give an update on the involvement of mitochondrial sirtuin SIRT2, SIRT3, SIRT4 and SIRT5 in chronic-degenerative disease focusing attention on their role in the development and progression of CRC. The manuscript is straightforward, well written, and concise and has clear results, within the scope of a review article. Definitely deserves to be published and is a valuable contribution to the “International Journal of Molecular Sciences” journal. Some minor comments need to be addressed before publication.

Response: We thank the Reviewer for helpful suggestions which undoubtedly allowed us to improve the manuscript.

Minor points:

[1] “1. Introduction”, Page 2 of 23, Lines 72-92:

“PPARÉ£ regulates metabolism at different levels and it has been observed that its reduced activity determined by microRNAs (miRNAs) activity or promoter hypermethylation can be found in CRC [14].”.

At that point the authors should highlight the potential utility of miRNAs as biomarkers in either tissues or blood for the assessment of response to the agents implemented in CRC, including the 5-fluorouracil based therapies, and EGFR inhibitors. They regulate critical pathways involved in the CRC pathogenesis, including the p53, PI3K, RAS, MAPK, EMT transcription factors, and Wnt/β-catenin pathways. Development of miRNA signature for predicting treatment response designates a personalized therapeutic approach of CRC.

Recommended reference: Boussios S, et al. The Developing Story of Predictive Biomarkers in Colorectal Cancer. J Pers Med. 2019;9(1):12.

Response: As suggested, we discussed the potential utility of miRNAs as biomarkers in CRC under Introduction section. “MiRNAs are single stranded molecule playing a crucial role in regulating gene expression. Several studies showed that they can be used as biomarkers in diagnostic, prognostic and therapeutic fields in CRC [14]. In this regard, miR-21 is an important biomarker for CRC early diagnosis and negative predictive factor for therapeutic response to 5-fluorouracil (5-FU), while miR-31 and miR-100 expression levels are positively correlated to resistance to Cetuximab-based therapy [14], highlighting as the development of miRNA signature for predicting treatment response can provide a personalized therapeutic approach of CRC”.

Line 76-84.

The recommended reference has been added as new ref 14.

[2] “3. Sirtuins and cancer”, Page 5 of 23, Lines 224-227:

“SIRT5 affects prostate cancer proliferation, progression and survival enhancinga cetyl-CoA acetyltransferase 1 activity leading to MAPK pathway promotion and ultimately to Matrix metallopeptidase 9 (MMP-9) and cyclinD1 expression [64].”

Cyclin D1 plays an important role in cell cycle progression through the association with CDK12. This is important as it has been reported that CDK12 mutations may delineate an immuno-responsive subgroup of prostate cancer with increased levels of T-cell infiltration and neoantigens. Based on that, CDK12-mutated tumors might constitute a separate subgroup of prostate cancer in which immunotherapy may be effective.

Recommended reference: Ghose A, et al. Genetic Aberrations of DNA Repair Pathways in Prostate Cancer: Translation to the Clinic. Int J Mol Sci. 2021;22(18):9783.

Response: As suggested, we discussed the role of CDK12 in prostate cancer development under “Sirtuins and cancer” paragraph; “An important role in cell cycle is carried out by Cyclin-dependent kinase 12 (CDK12), which can be found mutated in a subgroup of prostate cancer patients with infiltration of CD4+FOXP3- T-cells and neoantigens exposition, thus resulting in exalted immunotherapy response [66]. Therefore, CDK12-mutated tumors might constitute a separate subgroup of prostate cancer in which immunotherapy may be effective.”

Line 240-245.

The recommended reference has been added as new ref 66

[3] “5. Conclusions and future perspectives”, Pages 11 and 12 of 23, Lines 466-468:

“BZD9L1, a SIRT1/2 inhibitor, shows in vitro anti-cancer activities in HCT116 cells, subsequently to regulation of major cancer pathways and apoptosis induction through p53 [131].”.

This should be highlighted as TP53 protein, encoded by the TP53 tumor suppressor gene, has a regulatory role in cell growth arrest, DNA repair and apoptosis but also in oxidative stress, DNA damage and cell aberrant proliferation, thus maintaining cell cycle homeostasis. Mutated TP53 is expressed in about 60% of CRC. Point mutation in codon 72 - resulting in the substitution of proline to arginine - leads to dysfunction of the cell cycle “gatekeeper” that promotes the malignant process.

Recommended reference: Zarkavelis G, et al. Current and future biomarkers in colorectal cancer. Ann Gastroenterol. 2017;30(6):613-621.

Response: As suggested, in the Conclusion section, the role of TP53 protein has been highlighted. “BZD9L1, a SIRT1/2 inhibitor, shows in vitro anti-cancer activities in HCT116 cells, subsequently to regulation of major cancer pathways and apoptosis induction through p53, one of the most important regulators of cell cycle and DNA repair [138]. P53 acts as a transcription factor with tumor suppressor function and has been found mutated in codon 72 in a great percentage of CRC samples, promoting the acquisition of malignant properties [139].”

Line 503-508.

The recommended reference has been added as new ref 139

Reviewer 2 Report

In the review article by Colloca et al. the authors aimed to comprehensively review the latest advances about the role of mitochondrial sirtuins in affecting the development and progression of CRC. 

The topic is of great clinical importance, since SITRs may represent critical targets of newly developed anti-tumor therapies.

However, the review needs major revision regarding several aspects: 

  • though the 2nd part of the review focuses on mitochondrial SIRTs, in the INtroduction the bifunctional role of SITR1 in tumor development must be mentioned.
  • Also, in the Introduction, the classification of the human SIRT family must be mentioned.
  • Regarding the complex connections between SIRTs and cancer, their role in epithelial-to-mesenchymal transition, their possible connection with RTKs' signaling pathways (IGF1R, HGFR, EGFR), the complex interaction of SIRTs and autophagy must be discussed in detail. 

Several typos can be found in the text, that also need corrections.

After major revision, I suggest to accept the manuscript for publication. 

Author Response

Reply to Reviewer comments

Manuscript ID: ijms-1634624

Mitochondrial sirtuins in chronic degenerative diseases: new metabolic targets in colorectal cancer

Reviewer 2

Comments and Suggestions for Authors

In the review article by Colloca et al. the authors aimed to comprehensively review the latest advances about the role of mitochondrial sirtuins in affecting the development and progression of CRC.

The topicis of great clinical importance, since SITRs may represent critical targets of newly developed anti-tumor therapies. However, the review needs major revision regarding several aspects:

Response: We thank the Reviewer for helpful suggestions which undoubtedly allowed us to improve the manuscript.

-though the 2nd part of the review focuses on mitochondrial SIRTs, in the INtroduction the bifunctional role of SITR1 in tumor development  must be mentioned.

Response: As suggested, we discussed the bifunctional role of SIRT1 in tumor development under “Sirtuins and cancer” section; “SIRT1 shows an astonishing duality in cancer. In CRC, SIRT1 acts as an oncogene and can be found upregulated in conditions of depletion of SMAD Specific E3 Ubiquitin Protein Ligase 2 (SMURF2) [81]. The upregulation of SIRT1 by for Forkhead Box Q1 (FOXQ1) con-tributes to CRC survival inducing cancer cell radio-resistance and cell stemness [82] SIRT1, consequently to its phosphorylation and stabilization through c-Jun N-terminal kinases (JNK), favors CRC development in HCT116 cells through deacetylation of Snail protein and the subsequent upregulation of the expression of interleukin-6 and interleu-kin-8 [83]. On the other hand, SIRT1 can suppress tumor initiation increasing genome stability and inhibiting inflammation at the pre-cancer stage. Overexpression of SIRT1 can induce cell cycle arrest via E2F Transcription Factor 1 (E2F1) and represses CRC prolifera-tion and tumor initiation [70]. Sirtuins play a major role in cancer through the modulation of several autophagy-related pathways thus eliminating unnecessary or dysfunctional in-tracellular components. Particularly, SIRT2 promotes human CRC cell growth deacety-lating FOXO1 which is impaired in its binding to autophagy-related gene 7(ATG7), coun-tering autophagy-induced apoptosis [84].”

Line 282-293.

“SIRT1 promotes EMT in prostate cancer upregulating Snail and Twist or inhibiting of the expression of E-cadherin [70]”.

Line 254-256.

-Also, in the Introduction, the classification of the human SIRT family must be mentioned.

Response: As suggested, we provided the the classification of the human SIRT family under Introduction section. “Sirtuins can be divided into four classes I to IV: SIRT1, SIRT2 and SIRT3 belong to class I, SIRT4 to class II, SIRT5 to class III, and SIRT6 and SIRT7 to class IV [1]”.

Line 49-50.

-regarding the complex connections between SIRTs and cancer, their role in epithelial-to-mesenchymal transition, their possible connection with RTKs' signaling pathways (IGF1R, HGFR, EGFR), the complex interaction of SIRTs and autophagy must be discussed in detail.

Response: As suggested, the connections between SIRTs and cancer, their role in epithelial-to-mesenchymal transition, their possible connection with RTKs' signaling pathways, as well as complex interaction of SIRTs and autophagy have been discussed along “Sirtuins and cancer” and “Conclusion” sections;

In MCF-7 breast cancer cells, SIRT7 inhibition promotes stress resistance, downregulates insulin receptor (INSR), and modulates insulin-like growth factors (IGFs) pathways affecting cell metabolism and growth [61].”  

Line 226-228.

Recent evidence highlighted the antitumorigenic action of SIRT6 in CRC. SIRT6 indirectly opposes the intracellular lipid droplets (LDs) formation. Indeed, while LDs sustain human CRC cell metabolism and growth, SIRT6, activated by the FOXO3 transcription factor, counteracts their storage impairing lipid synthesis. The increase in LDs density depends on the cellular signal derived from the binding of epidermal growth factor (EGF) to its receptor EGFR that causes the inhibition of FOXO3/SIRT6 axis and consequently leads to a strongly increase in LDs production [78].

Line 274-280.

Sirtuins play a major role in cancer through the modulation of several autophagy-related pathways thus eliminating unnecessary or dysfunctional intracellular components. Particularly, SIRT2 promotes human CRC cell growth deacetylating FOXO1 which is impaired in its binding to autophagy-related gene 7(ATG7), countering autophagy-induced apoptosis [84].”

Line 293-297.

Recent studies described UBSC039 as a new SIRT6 activator. In several CRC cell lines, the activation of SIRT6 leads to enhanced ROS production, followed by an increased autophagy mediated by Beclin1 and autophagy-related gene5 (ATG5) [142].

Line 512-515.

-Several typos can be found in the text, that also need corrections.

Response: As suggested, manuscript has been corrected throughout for typos.

Round 2

Reviewer 2 Report

The revised version of the manuscript is now acceptable for publication. Congratulations to the authors.